# Investigating the Trade-off Between Accuracy and Theoretical Energy in Sparse ANN-to-SNN Conversion

## Abstract

Artificial Neural Networks (ANNs) are becoming increasingly important but face the challenge of the large scale and high energy consumption. Dynamic Sparse Training (DST) aims to reduce the memory and energy consumption of ANNs by learning sparse network topologies, which ultimately results in structural connection sparsity. Meanwhile, Spiking Neural Networks (SNNs) have attracted increasing attention due to their biological plausibility and event-driven nature, which ultimately results in temporal sparsity. To bypass the difficulty of directly training SNNs, converting pre-trained ANNs to SNNs (ANN2SNN) is becoming a popular approach to obtain high-performance SNNs. Here for the first time, we investigate the advantage of dynamically sparsely trained ANNs for conversion into sparse SNNs. By adopting Cannistraci-Hebb Training (CHT), a state-of-the-art brain-inspired DST family that resembles synaptic turnover during neuronal connectivity learning in brain circuits, we investigated the extent to which connectivity sparsity impacts the accuracy and theoretical energy efficiency of SNNs across different conversion approaches. The results show that sparse SNNs can achieve accuracy comparable to or even surpassing that of dense SNNs. Moreover, sparse SNNs can reduce theoretical energy consumption by up to 99% compared with dense SNNs. Furthermore, driven by the interest in understanding the physical dynamics interactions between firing rate and accuracy in SNNs, we systematically analyzed the temporal relationship between the saturation of firing rate and accuracy in SNNs. Our results reveal a significant time lag in which firing rate saturation precedes accuracy saturation. We also demonstrate that the magnitude of time lag is significantly different between sparse and dense networks, where the average time lag of sparse SNNs are higher than dense SNNs. Together, these results demonstrate that Cannistraci-Hebb Training can be effectively integrated into ANN-to-SNN conversion pipelines to obtain SNNs with competitive trade-off between accuracy and theoretical energy.

## 1 Introduction

Artificial neural networks (ANNs) have become central to various scientific, industrial, and economical applications. However, the high memory and energy costs of fully-connected ANNs create an urgent need for network architectures that consume much lower energy but still deliver comparable performance. One promising direction to reduce energy consumption is to sparsify a neural network(Blalock et al., 2020). Dynamic sparse training (DST), where both weights and topology evolve during training, has shown promises to discover sparse topology with close-to-dense performance and fewer links on ANNs(Mocanu et al., 2018; Jayakumar et al., 2020; Evci et al., 2020; Yuan et al., 2021; Zhang et al., 2024b). On hardware that supports sparse computation(Mishra et al., 2021; Dai et al., 2024), sparse networks require less computation, which translate the structural connection sparsity into energy savings(Tmamna et al., 2024).

Spiking Neural Networks (SNNs) are networks consisting of brain-inspired spiking neurons. Unlike neurons in ANNs, spiking neurons process information in time series with event-driven spikes(Eshraghian et al., 2023). Because of the event-driven nature of SNNs(Modaresi et al., 2023) and temporal sparsity of activations, SNNs thus hold promises to be far more energy-efficient than conventional ANNs on neuromorphic hardware(Davies et al., 2018; Merolla et al., 2014; DeBole et al., 2019; Song et al., 2022; Huo et al., 2023; Zhang et al., 2024a). In practice, however, training high-performance SNNs directly remains challenging due to the non-differentiablity of spiking neurons(Li et al., 2024), which has motivated alternative strategies for obtaining performant SNNs.

One popular option is ANN2SNN conversion: train a high-performance ANN with conventional methods and convert it to SNN. Conversion bypasses difficulties of direct training by utilizing well-established ANN training method. SNN conversion often yields near-ANN accuracy at low latency(Li et al., 2021; Bu et al., 2023; Huang et al., 2024; Wang et al., 2025), while being much more energy-efficient thanks to its event-driven nature.

However, to the best of our knowledge, prior ANN2SNN conversion works have focused most exclusively on dense networks(Li et al., 2021; Bu et al., 2023; Huang et al., 2024; Wang et al., 2025), while conversion on dynamically sparsely trained networks have never been studied. This gap is important because introducing structural connection sparsity into SNN conversion could combine benefits from both sides: the event-driven nature of SNNs reduces computation at temporal level, and introducing sparsity into structural connection can further reduce computation at structural level. Thus here, we extensively explored this intersection.

In this article we adopt established ANN2SNN algorithms(Yang et al., 2025; Wang et al., 2022; Chen et al., 2024; You et al., 2024) to investigate the extent to which introducing structural sparsity into these methods could influence the accuracy and theoretical energy consumption. Cannistraci-Hebb Training(CHT) as a brain-inspired, network-science-driven dynamic sparse training family, could learn not only weight but also topology of the networks, achieving close or even superior performance compared to the dense counterparts in ANNs(Zhang et al., 2024b; 2025; Hanming et al., 2025). Here we employ 3 sparse ANN structures, MLP(Rumelhart et al., 1986), VGG-16(Yu et al., 2021) and Vision Transformer base model(ViT-B)(Dosovitskiy, 2020; Touvron et al., 2021) derived with CHT. Then we conduct ANN2SNN conversion on pretrained ANNs through: three representative conversion approaches for MLP and VGG-16——CS-QCFS(Yang et al., 2025), SNM(Wang et al., 2022), and AEC(Chen et al., 2024)——and one representative conversion approach for Vision Transformer——SpikeZIP-TF(You et al., 2024).

Our results show that, across all conversion methods and ANN structures, sparse SNNs can achieve close or even superior accuracy compared to their dense counterparts. More importantly, regardless of the SNN conversion method, sparse SNNs can achieve a remarkable theoretical energy reduction(as large as 99%).

Driven by the interest in investigating the underlying mechanism of how SNNs achieve high-performance while consuming less energy theoretically compared with ANNs, we quantitatively studied the temporal relationship in the aspect of saturation between SNNs' accuracy and firing rate where firing rate directly influences the theoretical energy consumption. The saturation of dense SNNs's accuracy is commonly observed(Song et al., 2022; Han et al., 2020; Jiang et al., 2023; Bu et al., 2022; You et al., 2024). However, the quantitative relationship between saturation time of accuracy and firing rate has never been studied in current literature. Our results show that there exists a significant phenomenon that in SNNs firing rate saturate before accuracy. What is more interesting is that there also exists a significant difference in positive time lag between sparse and dense networks, which could be a potential cause of trade-off between accuracy and theoretical energy in sparse SNNs.

Together, these investigations position sparse ANNs derived from Cannistraci-Hebb Training as a useful component in developing efficient sparse ANN-to-SNN conversion pipelines, offering both practical computational benefits and a clearer understanding of the temporal dynamics of converted dense and sparse SNNs.

## 2   Methods

In this section we first introduce methods related to sparse ANN training and sparse SNN conversion. Please see Appendix A for details of 4 conversion methods(method 1,2,3 are for MLP and CNN; method 4 is for Transformer). Secondly, the theoretical energy measurement for SNNs and the principle of theoretical energy-saving by sparsity is explained. Thirdly, we demonstrate how to judge the saturation point. Finally, we present our experiment setup.

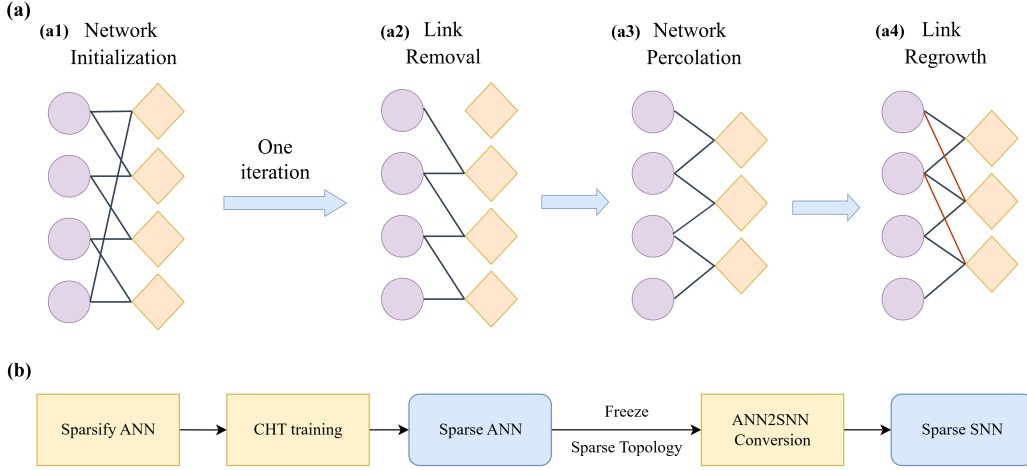

Figure 1: Overview of employed methods. (a) Illustration of CHT. (a1) Initiallize sparse network. (a2) Link removal. (a3) Network percolation, i.e. remove inactive neurons and paths. (a4) Link regrowth according to CH rule. (b) experiment pipeline for sparse models.

### 2.1   Sparse ANN training & sparse SNN conversion

#### 2.1.1   Cannistraci-Hebb Training

A sparse neural network is one neural network in which the number of connections between two layers is less than in a fully connected network of the same architecture. Cannistraci-Hebb Training (CHT) (Zhang et al., 2024b; Hanming et al., 2025; Zhang et al., 2025) is a network-driven and brain-inspired dynamic sparse training (DST) family that aims to learn the connectivity in networks. Here, sparsity level denotes the fraction of removed links compared with fully connected layer. During topology evolution a fraction of individual connections are canceled and new connections(generally, but not necessary, in the same amount of canceled) are grown.

Figure 1(a) presents the mechanism of CHT. First, a sparse network topology and weights are initialized[1]. For each topology evolution iteration during training, the procedure is:

  (i)  remove links according to their absolute weights.
  (ii)  remove inactive neuron and incomplete path (network percolation).
  (iii)  predict and grow new links using CHT network link prediction.

In this study, CHT soft rule (CHTs)(Zhang et al., 2025) is used on MLP and ViT-B, and CHT-Conv(Hanming et al., 2025) is used on VGG-16. The sparsity of linear layers in

---

[1]For MLP and CNN we train sparse ANNs from scratch; For ViT-B we adopt a hybrid structural-functional re-training pipeline for sparsify the dense model. First we initialize sparse ANN by pruning a pre-trained dense ANN to 70% sparsity according to absolute weight magnitude. Then we use CHT to finetune ViT-B to learn both the weight and topology.

MLP is 99%, sparsity of linear layers in ViT-B is 70%(Zhang et al., 2025), and sparsity of convolution layers in VGG-16 is 50%. Note that the output layer of models for classification should not be sparsified, so overall sparsity will be a bit lower than these values.

### 2.1.2 Sparse SNN conversion methods adaption

To adapt existing SNN conversion methods which are originally designed for dense SNN into sparse SNN, we do following adaption (see Figure 1(b)): First a sparse ANN is trained using CHT, then we record its topology and then convert the pre-trained sparse ANN to SNN. Topology of sparse SNN is frozen during the conversion.

### 2.2 Theoretical energy calculation of sparse and dense SNNs

On event-driven neuromorphic hardware, a SNN layer's forward computation is triggered only when spikes are emitted, which is the source of energy consumption(Eshraghian et al., 2023; Modaresi et al., 2023). In SNN the spikes are usually temporally sparse, which makes SNN more energy efficient than ANN.

The other important factor of SNN's energy efficiency energy is flops type. Because the spikes are binary, computation needed to forward spike to next layer is only AC(Accumulation), which is much cheaper than MAC(Multiply-and-Accumulate) used in traditional ANNs(Yao et al., 2023). Note that, for the input layer, if one uses Direct Input Encoding (DIE)(Rathi & Roy, 2021): the static image is fed to the SNN for $T$ times (thus the so called 'spike' is no more binary), the first layer's operations still should be deemed as MAC.

In this work we take $E_{\mathrm{MAC}} = 4.6\,\mathrm{pJ}$, $E_{\mathrm{AC}} = 0.9\,\mathrm{pJ}$ as the theoretical energy consumption of single MAC and AC operation(Yao et al., 2023).

Therefore, the theoretical energy consumption of a Spiking Neural Network should be

$$E = (total\ spikes) \times E_s \tag{1}$$

where $E_s$ is the theoretical energy consumption of single 'spike', and *total spikes* represents the total number of spikes in synapses in the network.

Note that in this study every 'energy' in Results/Appendix refers to the average theoretical energy consumption of SNN to infer one image on test set.

### 2.3 Saturation phenomenon analysis

#### 2.3.1 Model Average Spike Firing Rate

With inference time steps $T$ increasing, SNN properties such as accuracy(Song et al., 2022; Han et al., 2020; Jiang et al., 2023; Bu et al., 2022; You et al., 2024) and neuron firing rate tend to saturate. We introduce the concept of Model Average Spike Firing Rate(MASFR) as the firing rate averaged over all neurons in the model to examine the saturation of neuron firing rate in SNNs:

$$\mathrm{MASFR}(T) = \frac{\sum_{\mathrm{neuron} \in model} \sum_{t=1}^{T} \boldsymbol{spike}_{\mathrm{neuron}}^{l}(t)}{T \cdot N_{model}} \tag{2}$$

where $N_{model}$ is the number of neurons in the model.

#### 2.3.2 Algorithm to identify saturation time

Saturation means when $T$ is larger than a certain point, increasing $T$ doesn't bring significant improvement but instead converges to a stable value with some noise.

In this study, we used following method, which has shared concept with early stop, to determine the saturation time: If the relative improvement between time steps is continuously

no greater than 1% over 10 time steps, then the current time is determined as saturation point.

## 2.4 Experiment setup

In this experiment, we adopt MLP, VGG-16 (keep only one fully-connected layer to adapt for datasets no as challenging as ImageNet, see Yu et al. (2021)) and Vision-Transformer Base model (ViT-B, 16×16 patch size, 224×224 input) network structures. We train MLP and VGG-16 on popular image classification datasets CIFAR-10 and CIFAR-100(Krizhevsky et al., 2009). While ViT-B was adopted on ImageNet-1K(Deng et al., 2009) classification task. During sparse/dense ANN training and ANN2SNN conversion, grid-search is performed to obtain the best-performing ANNs and SNNs(except Vision Transformer) whose performances are shown in Results 3. The grid search spaces are present in Appendix B.

## 3 Results

In this section we present SNN results obtained by CHT and consequent SNN conversion. We also compared the SNNs converted from CHT trained ANN and pruned ANN, and we further compared SNN obtained by converting CHT trained ANN and STBP(Wu et al., 2018) sparse training(Han et al., 2024; 2025). Please refer to Appendix C and Appendix D respectively.

### 3.1 SNN accuracy analysis

SNN accuracy is related to its inference time steps $T$ where as $T$ increases, more spikes are emitted which stabilizes inference and makes inference accuracy gradually converge . Figure 2 plots how SNN accuracy changes with $T$. Note that for method 1,2,4(Yang et al., 2025; Wang et al., 2022), spiking neurons operate integration and firing at every time step, therefore we have inference time steps in $ts = [1, 2, 3, ..., 64]$ for Method 1,2,4. However, for method 3(Chen et al., 2024) the integration and firing operation is done in two separate time windows instead of at every time step. So on method 3 we treat $T$ as time window size which is a hyper-parameter. We display results with different $T \in ts = [2, 4, 8, 16, 32, 64]$ .

The first and second row in Figure 2 present results of CIFAR10 and CIFAR100 trained on MLP. As can be seen here, on both datasets, sparse ANNs can achieve a much higher accuracy than dense ANNs, showing the superiority of CHT training on ANNs. Also, comparing the accuracy between sparse and dense SNNs across 3 conversion methods, it could be observed that this accuracy advantage can be well preserved in sparse SNNs, which means on SNNs sparse networks can consistently achieve higher accuracy than the dense ones.

Third and Fourth row in Figure 2 present experiments using VGG-16 on CIFAR10 and CIFAR100, and the last row in Figure 2 present experiments of ViT-B on ImageNet. For VGG-16(on CIFAR10 and CIFAR100 datasets) and ViT-B(on ImageNet dataset), sparse ANNs derived by CHT have similar accuracies with dense ANNs. After conversion, sparse SNNs can achieve close or even superior performance compared to its dense counterparts.

Moreover, it could be observed that in method 1,2,4, there is no clear difference between the saturation time of sparse and dense networks. This means although with less links in the network, sparse SNNs can have similar information-processing efficiency compared with dense SNNs.

Considering the impact of connectivity, the above results show that: sparse ANNs trained by CHT can match or even exceed dense ANNs. And After ANN2SNN conversion, sparse SNNs well preserve this advantage with similar inference latency compared with dense ANNs.

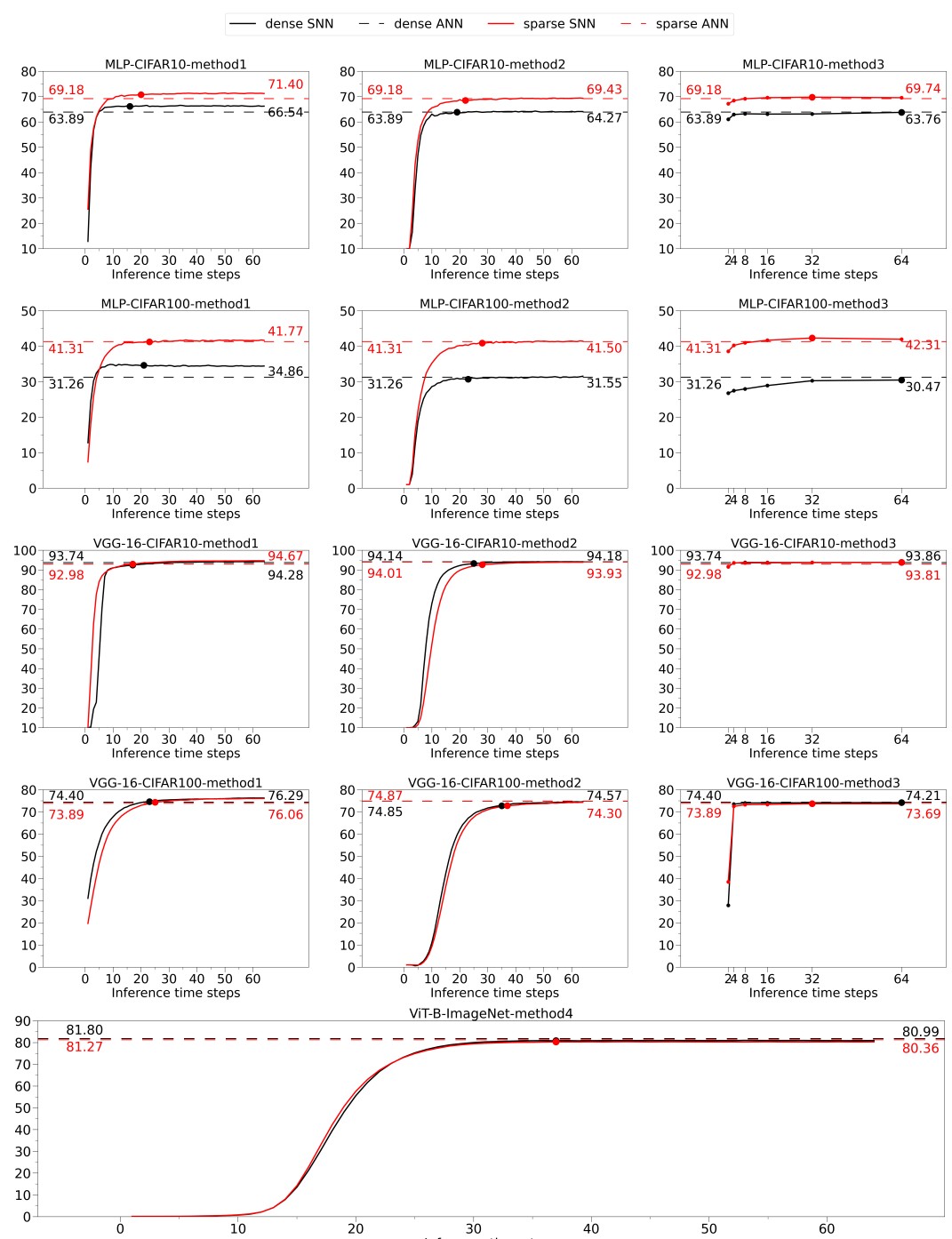

Figure 2: SNN accuracy(%) vs inference time steps. The accuracy curves in black are for dense models and red curves are for sparse models. While the solid lines represent how SNN model (after conversion) accuracy changes with time, dashed horizontal lines represent ANN (before conversion) accuracy. The point marked on the curve indicates the time: when SNN accuracy saturates (methods 1,2,4) / where integrate-and-fire window size has highest accuracy(method 3). For MLP and CNN, each row corresponds to a combination of network architectures and datasets while each column correspond to different ANN2SNN conversion methods. While in the last row, we show the results of ViT-B where we use ImageNet as benchmark dataset and method4 as conversion method. The black/red numbers near the left y-axis report the dense/sparse ANN accuracy; the black/red numbers at the right report the maximum dense/sparse SNN accuracy shown in the figure.

## 3.2 SNN theoretical energy consumption and accuracy analysis

In this section we analyze the theoretical energy consumption of dense and sparse SNNs according to Equation 1. Specifically for method 1,2,4, we assign time $T$ as the saturation time of SNN accuracy (see Section 2.3.2), while for method 3, $T$ denotes the time steps where SNN reach maximum accuracy. Using $T$ as reference time point, we calculate its theoretical energy consumption. The reason for assigning saturation time (for method 1,2,4) is a trade-off. If we assign $T$ smaller than saturation time, SNN's performance is not fully exploited. However, if we assign $T$ larger than saturation time, it will cause waste of energy because further inference time steps do not bring significant improvement in accuracy. Table 1 shows the theoretical energy and accuracy comparison between dense and sparse SNNs. For more detailed information about theoretical energy and accuracy, please refer to Appendix E.

Table 1: Theoretical energy consumption and accuracy comparison between dense and sparse SNNs. The extent of energy reduction is calculated as reduction $= \frac{E_{\text{dense}} - E_{\text{sparse}}}{E_{\text{dense}}} \times 100\%$. Accuracy improvement is calculated as sparse SNN accuracy minus dense SNN accuracy. In MLP, the sparsity level of linear layers is 99%; In VGG-16, the sparsity level of convolution layers is 50%; In ViT-B, the sparsity level of linear layers is 70%. Output layer are kept dense in all experiments.

| dataset | model | conversion method | energy reduction(%) | accuracy improvement(%) |
|---------|-------|-------------------|---------------------|-------------------------|
| CIFAR10 | MLP | QCFS | 99.05 | +4.13 |
| | | SNM | 98.83 | +4.63 |
| | | AEC | 99.16 | +5.98 |
| | VGG-16 | QCFS | 31.79 | +0.51 |
| | | SNM | 35.51 | -0.61 |
| | | AEC | 47.24 | -0.05 |
| CIFAR100 | MLP | QCFS | 99.03 | +5.79 |
| | | SNM | 98.72 | +10.17 |
| | | AEC | 98.63 | +11.84 |
| | VGG-16 | QCFS | 33.34 | -0.28 |
| | | SNM | 41.14 | +0.03 |
| | | AEC | 45.24 | -0.52 |
| ImageNet | ViT-B | SpikeZIP-TF | 58.87 | -0.48 |

Results show regardless of architectures, datasets or conversion methods, sparse SNNs are always more energy-efficient than dense SNNs theoretically. That is because sparse SNNs benefit from structure connection sparsity that reduces active links compared with dense ANNs.

Therefore, regardless of conversion method, for MLP with sparse linear layers with 99% of sparsity, sparse SNNs save up to 99% of energy theoretically; the smallest observed reduction 98.63% is still incredible. For CNN with sparse convolution layers with 50% of sparsity, sparse SNNs achieve theoretical energy reduction is always higher than 30%. For ViT with sparse lineary layers with sparsity 70%, sparse SNNs can save 58.87% of theoreticalenergy compared with dense SNNs.

The last column presents the accuracy improvement of sparse SNNs relative to dense SNNs. In 8 out of 13 experiments, sparse SNNs achieve not only theoreticalenergy reduction but also accuracy improvement. In other cases, sparse SNNs have comparable accuracy with its dense counterparts, but reduce theoretical energy consumption to a large extent.

### 3.3 Time lag between saturation of Firing rate and Accuracy

For a deeper investigation of the underlying mechanism of how spikes firing rate affects model performance in SNNs, we here systematically studied the relationship between saturation time of accuracy and Model Average Spike Firing Rate (MASFR) in SNNs, where saturation time is calculated using the same algorithm described in Section 2.3.2. Our analysis utilizes data from all grid-search experiments involving methods 1,2 across four architecture-dataset combinations, as in method 1,2 the integration and firing operations and accuracy updates happen at every time step. By incorporating a wide range of methods, architectures, datasets, and hyper-parameter configurations, we aim to understand the general dynamics of SNNs obtained through ANN2SNN conversion.

Figure 3(a) presents a scatter plot of MASFR saturation time versus accuracy saturation time for dense (left panel) and sparse (right panel) SNNs. Time lag is defined as

$$\text{time lag} = \text{accuracy saturation time} - \text{MASFR saturation time} \tag{3}$$

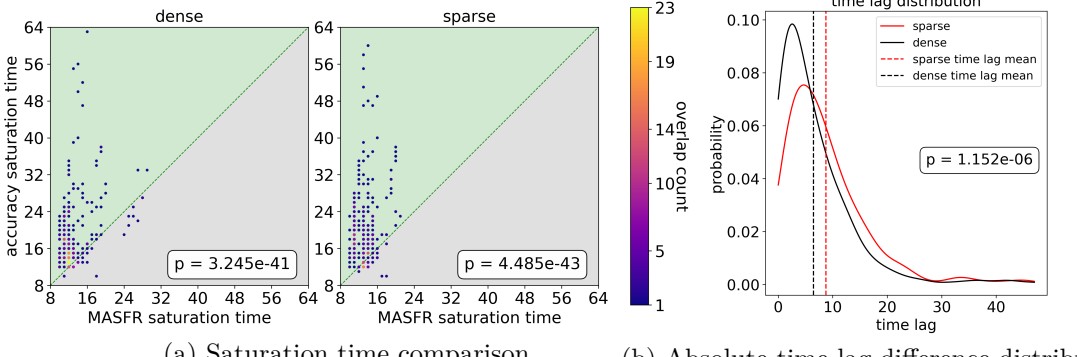

(a) Saturation time comparison     (b) Absolute time lag difference distribution

Figure 3: Time lag analysis. (a) Points located in the green-shaded region above the y=x line indicate that firing rate saturates earlier than accuracy. The colors of the points correspond to the number of overlapping data points. (b) The distribution of time lag of dense and sparse SNNs. The vertical dashed lines represent the mean time lag value. Red curves are for sparse SNNs and black curves are for dense SNNs.

To statistically evaluate the significance of this phenomenon, a one-sided Wilcoxon signed-rank test was conducted between the saturation time of accuracy and MASFR. The results are: for dense SNNs $p$-value $= 3.245 \times 10^{-41}$, sparse SNNs $p$-value $= 4.485 \times 10^{-43}$, and all SNNs $p$-value $= 3.865 \times 10^{-82}$. These results provide strong statistical evidence that MASFR saturation precedes accuracy saturation. Given the diversity of the experimental

settings, this conclusion suggests that the observed time lag is a general characteristic of SNNs.

This phenomenon can be qualitatively understood from the perspective of rate decoding(used in method 1,2). The sufficient condition for accuracy saturation is the stabilization of firing rates in the output layer neurons. Since MASFR is an average across all neurons in the network, it takes additional time for firing rate of neurons in the last layer to stabilize after MASFR's saturation. Thus there is a time lag between saturation of accuracy and MASFR.

After recognizing that positive time lag is significantly dominant, we further focused on studying the difference between positive time lag of sparse SNNs and positive time lag of dense SNNs. The distribution of time lag of two connectivity types are plotted using Kernel Density Estimation in Figure 3(b). A two-sided Mann-Whitney test between time lag of sparse SNNs and time lag of dense SNNs shows $p$-value $= 1.152 \times 10^{-6}$, which gives solid evidence to the difference in time lag between sparse SNNs and dense SNNs. Moreover, the mean value of time lag of sparse SNNs is higher than that of dense SNNs. This is a very interesting phenomenon because it shows how structural connectivity impacts the SNN mechanism.

To summarize, this study uncovers a phenomenon of converted SNN using rate coding: the time lag between accuracy and MASFR. Moreover, the time lag of sparse SNNs are significantly different from the time lag of dense SNNs, while the average time lag of sparse SNNs are higher than dense SNNs. This may be a potential cause of the accuracy and theoretical energy advantage of sparse SNNs over dense SNNs.

## 4 Discussion

Here for the first time, we perform the investigation on the sparse SNNs converted with dynamically sparsely trained ANNs. We show that sparse SNNs can achieve close and in some cases superior accuracy compared to their dense counterparts while being substantially more energy-efficient, making a theoretical energy reduction up to 99%. This introduces an efficient conversion pipeline which can achieve good trade-off between accuracy and theoretical energy in sparse spiking neural networks, which is not obtainable through either ANN-to-SNN conversion or dynamic sparse training alone. The reason for this superior performance in sparse networks is that during Cannistrci-Hebb training, various topological properties critical to efficiency of a network such as low characteristic path length and hyperbolic community structure start to emerge.(Zhang et al., 2024b) Also, sparsity in networks adds more non-linearity in learning, thus enabling the model to learn a better representation of features. By transferring the sparse advantage of Cannistraci-Hebb Training into spiking neural networks, this work achieves a competitive trade-off between accuracy and theoretical energy in sparse SNNs.

Additionally, to the best of our knowledge, the relationship between firing-rate saturation and accuracy saturation in sparse SNN have not been quantitatively analyzed in prior SNN conversion work. Our study reveals a consistent positive time lag across methods and architectures, and a significant difference in this time lag between sparse and dense networks. This provides new insight into how sparse connectivity affects temporal information processing in converted SNNs, which might be a potential cause for the performance and the theoretical energy gap between networks with different sparsity level.

Despite its significance, this work has certain limitations. Limited by available hardware, we analyze theoretical energy consumption rather than measuring real energy consumption. Our theoretical energy calculation is based on future hardware with the support of both sparse and event-driven computation. Furthermore, a long inference time is observed in spiking neural networks converted using AEC method (method 3). The reason is that in the case of AEC, instead of performing integrate-and-fire operation in real-time, AEC do integrate and fire operation separately in two time windows, and in the experiments a detailed grid-search is performed on this window size (within 2,4,8,16,32,64) and the best-performing window size is shown in the main results. Under this circumstance, a larger

window size will allow longer integration-and-fire time for the model to process more spiking information, thus resulting in higher accuracy.

Looking ahead, there are several promising directions that could extend the results of this work. The first direction is to study how to use CHT to directly train SNNs so as to further evaluate the effectiveness of DST methods(such as CHT) on SNNs. What's more, future study can extend time lag phenomenon study to various architectures, datasets and methods to fully understand the SNN model and time dynamics. Last but not least, we suggest the development of sparse neuromorphic hardware, as it is promising to implement energy-efficient architectures in reality.

Reproducibility

Code of this research is sumbmitted as supplementary material. Please read README.md for more information.

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

# A Appendix: ANN2SNN conversion methods

To thoroughly examine how sparsity affects SNN, we choose three conversion methods with distinct principles: CS-QCFS, SNM, AEC and SpikeZIP-TF.

## A.1 Method1: CS-QCFS

CS-QCFS converts ANNs to SNNs by first replacing ReLU with a Softplus Quantization Clip-Floor-Shift (S-QCFS) activation function on pre-trained ANNs for further training to learn learn layer-wise thresholds. Then, S-QCFS function is replaced with channel-wise S-QCFS(CS-QCFS) function to finetune channel-wise thresholds in convolution layers, which aims to capture heterogeneous activation distribution across channels. Finally, the ANN weights are directly transferred to SNN. Channel-wise thresholds of CS-QCFS are used as SNN spiking threshold, after applying a soft-plus function to ensure positivity of SNN's threshold.

## A.2 Method2: SNM

Signed Neuron with Memory (SNM) allows both positive and negative spikes. If membrane potential exceeds positive threshold, neuron will emit a positive spike ($+1$). Despite this classical Integrate-Fire(IF) neuron behavior, for SNM neuron if membrane potential is lower than negative threshold, neuron will emit a spike ($-1$). SNM also includes a memory mechanism to make sure the number of positive spike is no less than number of negative spike, which ensures non-negative firing rate. Neuron-wise normalization is applied to determine the spiking threshold, in which for each neuron in a pre-trained ANN the maximum activation is set as the neuron's spiking threshold.

## A.3 Method3: AEC

AEC (At-Most-Two-Spike Exponential Coding) splits each layer's inference window into a decoding phase and an encoding phase. During decoding phase neurons only accumulate inputs from previous layer and stay silent. During encoding phase each neuron can emit up to two spikes: primary and compensate; A primary spike occurs once membrane potential (accumulated in decoding phase) exceeds primary threshold. A reset-by-subtraction is done after primary spike. Then a compensate spike occurs if the membrane potential after reset exceeds compensate threshold. Both primary and compensate threshold decays exponentially with time. The output of the layer is the product of thresholds(at spike time) and spikes. After replacing ANN neurons by AEC neurons with fixed initialization, a further training on SNN is applied to only finetune the threshold parameters and BatchNorm parameters. Finally in inference BatchNorm layers are folded into preceding convolution layer to yield output SNN.

## A.4 Method4: SpikeZIP-TF

SpikeZIP-TF is a novel ANN-to-SNN conversion framework designed to establish exact functional equivalence between quantized Transformer-based ANNs and Spiking Neural Networks (SNNs). This method follows a pipeline involving ReLU replacement, end-to-end finetuning, Quantization-Aware Training (QAT), and conversion using specialized spiking components.

Central to this approach is the ST-BIF+ neuron, which can emit negative spike additional to IF neuron. To handle the complex computation of Transformers, the authors propose Spike-Equivalent Self-Attention (SESA). SESA decomposes both Activation-Weight and Activation-Activation matrix multiplications into spike-based computations that can reconstruct quantized self-attention. Furthermore, the framework introduces Spike-Softmax and Spike-LayerNorm to handle SNN-unfriendly nonlinear operators.

## B   Appendix: Grid search space for ANN training and ANN2SNN conversion

TableA1 below shows the hyper-parameters and grid search spaces of ANN training and SNN conversion methods 1,2,3. lr represents learning rate, bs represents batch size. L is quantization steps for CS-QCFS method. For fine-tuning ViT after pruning we use emperical value according to Touvron et al. (2021), and for SNN conversion method 4 for ViT we use emperical value from You et al. (2024).

Regrading CHT configurations, we use CH2-L3n-soft for linear layers, and CHT-Conv for convolution layers.

Table A1: Hyper-parameters and grid search space of ANN training and ANN2SNN conversion methods

| ANN training | |
| --- | --- |
| lr | (0.1, 0.001, 0.0001) |
| bs | (32, 64, 128) |

| Method 1 | |
| --- | --- |
| lr | (0.05, 0.01, 0.005, 0.001, 0.0005) |
| bs | (32, 64, 128) |
| L | (2, 4, 8, 16, 32) |

| Method 2 | |
| --- | --- |
| bs | (32, 64, 128) |

| Method 3 | |
| --- | --- |
| lr | (0.001, 0.0001, 0.00001) |
| bs | (32, 64, 128) |

## C   Appendix: CHT vs. Pruning Under Matched Sparsity

In this section, we compare the network-wise sparsity, theoretical energy and accuracy of 2 SNN VGG-16 networks derived from CHT and an influential structural pruning method DepGraph(Fang et al., 2023) with similar sparsity, converted with QCFS method (Method 1). Looking at Table A2, the results suggests that across 2 different datasets CIFAR10 and CIFAR100, under similar sparsity, CHT has higher accuracy while achieving comparable theoretical energy consumption. Note that network-wise sparsity means the overall link sparsity of the model. This is especially obvious in CIFAR100, where CHT's accuracy is much higher than DepGraph (74.32% vs. 71.16%). Thus, the competitive trade-off between accuracy and theoretical energy doesn't come from mere sparsity level, but also depends on the method to achieve sparsity, illustrating the advantage of dynamic sparse training method.

Table A2: Comparison between sparse SNNs obtained by CHT and Depgraph pruning.

| Dataset | sparsification method | network-wise sparsity(%) | T | energy($\mu$J) | accuracy(%) |
| --- | --- | --- | --- | --- | --- |
| CIFAR10 | CHT | 44.29 | 17 | 244.14 | 92.97 |
| | Depgraph | 43.48 | 18 | 229.12 | 92.43 |
| CIFAR100 | CHT | 44.29 | 25 | 434.38 | 74.32 |
| | Depgraph | 43.48 | 25 | 344.90 | 71.16 |

## D   Appendix: CHT vs. STBP-Based SNN sparse training Methods

To compare the inference and energy consumption between sparse SNNs directly trained from STBP and converted from CHT-trained sparse ANNs, we selected 2 state-of-the-art STBP-based SNN sparse training methods(Wu et al., 2018) to compare with CHT, namely

SD-SNN(Han et al., 2025) and DPAP(Han et al., 2024). We adopt the same network structure as in SOTA articles, which is a convolutional network with 6 convolutional layers followed by 2 fully connected layers, and do dynamic sparse training on CHT to derive the sparse ANN, which is then converted to SNN with CS-QCFS (Method 1) for comparison. Table A3 shows a comparison on SNN accuracy and theoretical energy consumptions between different methods. The results suggest that across 2 CIFAR datasets, CHT networks achieves the highest network-wise sparsity (87.1%). Given same inference time steps (T=8), CHT consumes least energy theoretically and achieves comparable accuracies with STBP-based SNN sparse training methods.

Table A3: Comparison between sparse SNNs converted from CHT-trained-ANN and direct STBP-based sparse training.

| Dataset | method | network-wise sparsity(%) | energy($\mu$J) | accuracy(%) |
|---|---|---|---|---|
| CIFAR10 | DPAP | 57.87 | 488.54 | 94.21 |
| | SD-SNN | 35.26 | 368.92 | 94.32 |
| | CHT+CS-QCFS | 87.1 | 226.74 | 93.00 |
| CIFAR100 | DPAP | 57.75 | 601.61 | 74.5 |
| | SD-SNN | 37.02 | 426.31 | 74.86 |
| | CHT+CS-QCFS | 87.1 | 339.00 | 74.30 |

## E  Appendix: Detailed results of dense and sparse SNNs

Table A4 presents the detailed information of absolute theoretical energy, accuracy and time steps for dense and sparse SNNs.

## F  Appendix: LLMs usage

We used LLMs to polish the language of the article while the principle, main logic, and the innovations of the article are by the authors. We also used LLMs to search literature, but they are all verified and read by authors.

Table A4: Detailed results of dense and sparse SNNs. For each method, the first row is information about dense SNNs, and the second row is information about sparse SNNs.

| dataset | model | method | T | energy($\mu$J) | accuracy(%) |
|---|---|---|---|---|---|
| CIFAR10 | MLP | QCFS | 16 | 722.69 | 66.12 |
| | | | 15 | 6.84 | 70.25 |
| | | SNM | 19 | 859.46 | 63.85 |
| | | | 22 | 10.09 | 68.47 |
| | | AEC | 64 | 21.21 | 63.76 |
| | | | 32 | 0.18 | 69.74 |
| | VGG-16 | QCFS | 17 | 357.89 | 92.46 |
| | | | 17 | 244.14 | 92.97 |
| | | SNM | 25 | 887.58 | 93.27 |
| | | | 28 | 572.39 | 92.66 |
| | | AEC | 64 | 99.42 | 93.86 |
| | | | 64 | 52.45 | 93.81 |
| CIFAR100 | MLP | QCFS | 21 | 950.30 | 34.63 |
| | | | 20 | 9.25 | 40.42 |
| | | SNM | 23 | 1041.65 | 30.71 |
| | | | 28 | 13.32 | 40.88 |
| | | AEC | 64 | 20.38 | 30.47 |
| | | | 32 | 0.28 | 42.31 |
| | VGG-16 | QCFS | 23 | 651.68 | 74.60 |
| | | | 25 | 434.39 | 74.32 |
| | | SNM | 35 | 1201.14 | 72.76 |
| | | | 37 | 707.00 | 72.79 |
| | | AEC | 64 | 99.18 | 74.21 |
| | | | 32 | 54.31 | 73.69 |
| ImageNet | ViT-B | SpikeZIP-TF | 37 | 169715.88 | 80.78 |
| | | | 37 | 69800.39 | 80.30 |