# OpenReview forum: "Conversion of sparse Artificial Neural Network to sparse Spiking Neural Network can save up to 99% of energy"
_ICLR.cc/2026/Conference — Submitted to ICLR 2026_

### Official Review · Reviewer_3cJ8 · 2025-10-26

**Soundness:** 2
**Presentation:** 1
**Contribution:** 1
**Rating:** 2
**Confidence:** 5

**Summary:**

This work combines ANN-SNN Conversion framework with a Dynamic Sparse Training (DST) scheme named Cannistraci-Hebb Training (CHT) to achieve low-power SNNs.

**Strengths:**

1. This work explores the feasibility of obtaining low-power SNNs from the perspective of conversion learning rather than SNN direct training (or referred to as STBP training).

**Weaknesses:**

1. This work only shows the energy-saving ratio in Table 2, but does not simultaneously display the inference accuracy under the corresponding sparsity level. In addition, this work only presents two small-scale static datasets (CIFAR-10/100) and convolutional network structures.

2. The so-called energy saving in this work are compared to vanilla pretrained ANN models. This work did not compare the inference accuracy and energy consumption with a series of important works based on STBP sparse training.

3. The sparse SNN obtained based on ANN-SNN Conversion requires a significant amount of time-steps in the inference phase, as evidenced by the time-steps listed in Table 2. In comparison, the SNN obtained from STBP sparse training usually only requires no more than 8 time-steps, which is also a limitation of this work.

4. The sparse training in this work was conducted during the ANN stage and is not directly related to SNN, which raises concerns about the contribution of this work to the SNN community.

5. It is obvious that the layout of the figures, tables and formulas in this work needs further optimization.

**Questions:**

See Weaknesses Section.

---

> ### Author Response · Authors · 2025-12-03
>
> **NOTES FROM AUTHORS**
>
> 1) We have addressed the first major Reviewers’ concern on hyperparameter search. Now, the methodology for training dense ANN models is revised so that for dense and sparse ANNs, hyperparameter grid-search is conducted separately to find the best-performing models, instead of applying the best hyperparameters derived from sparse models on both dense and sparse networks.
> 2) We have addressed the second major Reviewers’ concern on computation of energy consumption in SNNs. Now, the energy is calculated by summing up the energy consumptions of all spikes on the synapses, instead of estimating energy based on average firing rates.
>
> **W1**: This work only shows the energy-saving ratio in Table 2, but does not simultaneously display the inference accuracy under the corresponding sparsity level. In addition, this work only presents two small-scale static datasets (CIFAR-10/100) and convolutional network structures.
>
> **REPLY TO W1**:
>
> We thank reviewer for the deep insight and kind suggestion.
>
> 1. We have restructured [Table 1](https://anonymous.4open.science/r/rebuttal-figure-table-5424/Table1.png)  (previously Table 2) to show not only the energy reduction but also the improvement in accuracy achieved by sparse SNNs compared to their dense counterparts. As can be seen, in 8 out of 13 groups of experiments, sparse SNNs can outperform the dense ones in accuracy, and the performances are comparable in other cases. And in all cases, sparse network can achieve a more than 30% reduction in energy, indicating a good trade-off between accuracy and theoretical energy.
>
> 2. Following your suggestion, to demonstrate the generalizability of the method proposed here, we further conduct experiments on Vision Transformer (ViT). We proposed a hybrid structural-functional re-training pipeline for sparsifying the dense ViT model. In brief: we took the already dense Vision Transformer-Base (ViT-B, 16×16 patch size, 224×224 input) model proposed by Dosovitskiy et al[1] pre-trained for ImageNet classification task; we applied one-shot absolute weight pruning at 70% sparsity followed by CHT re-training; we converted the sparse re-trained model to SNN to evaluate its performance on the challenging ImageNet classification task. The last row of [Figure 2](https://anonymous.4open.science/r/rebuttal-figure-table-5424/Figure2.png) shows the accuracy of dense and sparse ANN ViTs and how SNN ViTs’ accuracy evolve with inference time step. Here the black/red numbers near the left y-axis report the dense/sparse ANN accuracy; the black/red numbers at the right report the maximal accuracy of dense/sparse SNNs. It could be observed that:
>
>     (1)	Sparse ANN ViT trained with CHT achieves comparable performance with dense ANN ViT
>
>     (2)	After conversion, both sparse and dense SNNs achieves similar accuracy.
>
>     Furthermore, the last row in [Table 1](https://anonymous.4open.science/r/rebuttal-figure-table-5424/Table1.png)  compares the accuracy and theoretical energy consumption between sparse and dense SNN ViTs. It could be observed that:
>
>     (1)	The sparse SNN ViT can reduce 58.87% theoretical energy consumption compared to its dense counterpart with only 0.48% loss in accuracy.
>
>     These results confirm that the trade-off between accuracy and theoretical energy achieved by our method can also generalize to more complex network structures and more challenging tasks.
>
> [1] Dosovitskiy, Alexey. "An image is worth 16x16 words: Transformers for image recognition at scale." arXiv preprint arXiv:2010.11929 (2020).
>
>
> **SEE FURTHER REPLIES IN NEXT COMMENTS**

---

> ### Author Response · Authors · 2025-12-03
>
> **W2**: The so-called energy saving in this work are compared to vanilla pretrained ANN models. This work did not compare the inference accuracy and energy consumption with a series of important works based on STBP sparse training.
>
> **W3**: The sparse SNN obtained based on ANN-SNN Conversion requires a significant amount of time-steps in the inference phase, as evidenced by the time-steps listed in Table 2. In comparison, the SNN obtained from STBP sparse training usually only requires no more than 8 time-steps, which is also a limitation of this work.
>
> **REPLY TO W2&W3**:
>
> We thank reviewer for the constructive suggestion.
>
> 1) Regarding your concern on energy savings, the energy stats shown in [Table 1](https://anonymous.4open.science/r/rebuttal-figure-table-5424/Table1.png) (previously Table 2)   are all measured on SNN. We found this might be an oversight on table caption, now in the table caption we have clearly stated that the energy consumption and accuracies are measured on SNNs.
> 2) To compare the inference and energy consumption between sparse SNNs directly trained from STBP and converted from CHT-trained sparse ANNs, we selected 2 state-of-the-art STBP[2]-based SNN sparse training methods to compare with CHT, namely SD-SNN[3] and DPAP[4]. We adopt the same network structure as in SOTA articles, which is a convolutional network with 6 convolutional layers followed by 2 fully connected layers, and do dynamic sparse training on CHT to derive the sparse ANN, which is then converted to SNN with QCFS (Method 1) for comparison.
> [Table A3](https://anonymous.4open.science/r/rebuttal-figure-table-5424/TableA3.png) shows a comparison on SNN accuracy and energy consumptions between different methods. The results suggest that across 2 CIFAR datasets, CHT networks achieves the highest network-wise sparsity (87.1%). Given same inference time steps (T=8), CHT consumes least energy and achieves comparable accuracies with STBP-based sparse training methods. We now add this as a new section [CHT vs. STBP-Based SNN sparse training Methods] in the appendix.
>
>     We have also added a new section in Appendix named [CHT vs. Pruning Under Matched Sparsity], where we compare the network-wise sparsity, energy and accuracy of 2 SNN VGG-16 networks derived from CHT and an influential structural pruning method DepGraph[5] with similar sparsity, converted with QCFS method (Method 1), which is widely adopted. Looking at [Table  A2](https://anonymous.4open.science/r/rebuttal-figure-table-5424/TableA2.png), the results suggest that across 2 different datasets CIFAR10 and CIFAR100, under similar sparsity, CHT has higher accuracy while achieving comparable energy consumption. This is especially obvious in CIFAR100, where CHT’s accuracy is much higher than DepGraph (74.32% vs. 71.16%).
>
>     Thus, according to the results above, the trade-off between accuracy and theoretical energy does not derive from mere sparsity level, but also depends on the method to achieve sparsity, illustrating the role of dynamic sparse training method such as CHT.
>
> [2] Wu, Yujie, et al. "Spatio-temporal backpropagation for training high-performance spiking neural networks." Frontiers in neuroscience 12 (2018): 331.
>
> [3] Han, Bing, et al. "Adaptive sparse structure development with pruning and regeneration for spiking neural networks." Information Sciences 689 (2025): 121481.
>
> [4] Han, Bing, et al. "Developmental plasticity-inspired adaptive pruning for deep spiking and artificial neural networks." IEEE Transactions on Pattern Analysis and Machine Intelligence (2024).
>
> [5] Fang, Gongfan, et al. "Depgraph: Towards any structural pruning." Proceedings of the IEEE/CVF conference on computer vision and pattern recognition. 2023.
>
> **SEE FURTHER REPLIES IN NEXT COMMENTS**

---

> ### Author Response · Authors · 2025-12-03
>
> **W4**: The sparse training in this work was conducted during the ANN stage and is not directly related to SNN, which raises concerns about the contribution of this work to the SNN community.
>
> **REPLY TO W4**:
>
> Thank you for your comment. We agree that sparsity is introduced during the ANN training stage and that the SNN is obtained through a standard ANN-to-SNN conversion process. However, the present study makes two contributions that are directly relevant to the SNN community
>
> 1) We show that applying Cannistraci-Hebb sparse training before conversion can enable the sparse SNNs retain accuracy comparable or even higher than dense SNNs while achieving substantial theoretical energy savings. This introduces an efficient conversion pipeline which can achieve good trade-off between accuracy and theoretical energy in sparse spiking neural networks, which is not obtainable through either component alone.
>
> 2) To the best of our knowledge, both the conversion of dynamically sparsely trained ANN to SNN, and the relationship between firing-rate saturation and accuracy saturation in sparse SNN have not been quantitatively analyzed in prior SNN conversion work. Our study reveals (i) a consistent positive time lag across methods and architectures, and (ii) a significant difference in this time lag between sparse and dense networks. This provides new insight into how sparse connectivity affects temporal information processing in converted SNNs, which might be a potential cause for the performance and energy gap between networks with different sparsity level.
>
> To address the reviewer’s concern:
> We have revised the Discussion section of the article to include the discussion of this valuable point raised by the Reviewer.
>
> **W5**: It is obvious that the layout of the figures, tables and formulas in this work needs further optimization
>
> **REPLY TO W5**:
>
> Thank you for your kind suggestion. We now have restructured the tables in the manuscript to increase readability.
> Specifically, we moved the grid-search space of ANN training and 3 ANN-to-SNN conversion method (originally Table 1) to Appendix B as presented in
> [Table A1](https://anonymous.4open.science/r/rebuttal-figure-table-5424/TableA1.png).
> Furthermore, for [Table 1](https://anonymous.4open.science/r/rebuttal-figure-table-5424/Table1.png) (previously Table 2) , we removed redundant information for comparison, and only preserves the energy reduction and accuracy improvement achieved by the sparse SNNs compared to their dense counterparts. The detailed information of saturation time step (T), energy consumption, and accuracy are now moved to Appendix E as presented in
> [Table A4](https://anonymous.4open.science/r/reply2-5771/TableA4.png).

---

### Official Review · Reviewer_nyYp · 2025-10-30

**Soundness:** 2
**Presentation:** 2
**Contribution:** 2
**Rating:** 2
**Confidence:** 3

**Summary:**

This paper investigates whether dynamically sparse artificial neural networks (ANNs), trained using the Cannistraci-Hebb Training (CHT) algorithm, can improve the performance and energy efficiency of spiking neural networks (SNNs) when converted through existing ANN-to-SNN conversion methods. The authors report that converting dynamically sparse ANNs to sparse SNNs maintains comparable accuracy to dense baselines while achieving substantial theoretical energy reductions (up to 99%)

**Strengths:**

Originality.
Poses a concrete and timely question at the intersection of dynamic sparsity and ANN to SNN conversion: do topology-sparse ANNs yield accuracy/energy benefits after conversion when topology is preserved?

Quality.
Provides a clear experimental pipeline: train sparse ANNs via CHT, freeze topology, convert with three representative methods, then evaluate accuracy vs. time steps and theorized energy.

Clarity.
The paper is structured and readable (method figures, tables, and definitions are straightforward).

Significance.
If validated on hardware and broader settings, the claim that sparse-topology conversion preserves accuracy while dramatically reducing energy would matter for energy-aware neuromorphic deployment.

**Weaknesses:**

1. Novelty appears incremental: The study combines two well-established components, namely dynamically sparse training (DST)-based sparsity and standard ANN-to-SNN conversion techniques, and focuses mainly on evaluating their combined effect rather than introducing a new methodological contribution.

2. Energy claims are theoretical and hinge on strong assumptions; no hardware validation. The headline “up to 99% energy reduction” is derived from a spike-count/FLOP model plus constants (EMAC/EAC) and the assumption that sparse hardware gives linear speed/energy benefits w.r.t. link sparsity. There is no measurement on neuromorphic or sparse-compute hardware. Consequently, the core claim remains speculative without device-level latency/energy evidence or even cycle-accurate simulators.

**Questions:**

1. Hardware validation: Can you provide on-device latency and energy results for one platform to substantiate the 99% savings, and report how close real savings are to the theoretical model?
2. Scope expansion: Do results persist on larger datasets (e.g., Tiny-ImageNet/ImageNet-subset) and deeper/backbone variants (e.g., ResNet/transformers) to support broad claims?

---

> ### Author Response · Authors · 2025-12-03
>
> **NOTES FROM AUTHORS**
>
> 1) We have addressed the first major Reviewers’ concern on hyperparameter search. Now, the methodology for training dense ANN models is revised so that for dense and sparse ANNs, hyperparameter grid-search is conducted separately to find the best-performing models, instead of applying the best hyperparameters derived from sparse models on both dense and sparse networks.
> 2) We have addressed the second major Reviewers’ concern on computation of energy consumption in SNNs. Now, the energy is calculated by summing up the energy consumptions of all spikes on the synapses, instead of estimating energy based on average firing rates.
>
> **W1**: Novelty appears incremental: The study combines two well-established components, namely dynamically sparse training (DST)-based sparsity and standard ANN-to-SNN conversion techniques, and focuses mainly on evaluating their combined effect rather than introducing a new methodological contribution.
>
> **REPLY TO W1**:
>
> We appreciate the reviewer’s observation and the opportunity to clarify the novelty of our contribution. While it is true that our work builds upon established components—dynamic sparse training (DST) and ANN-to-SNN conversion—the contribution goes beyond a simple combination of these techniques. Our study provides two key contributions:
>
> 1) We show that applying Cannistraci-Hebb sparse training before conversion can enable the sparse SNNs retain accuracy comparable or even higher than dense SNNs while achieving substantial theoretical energy savings. This introduces an efficient conversion pipeline which can achieve good trade-off between accuracy and theoretical energy in sparse spiking neural networks, which is not obtainable through either component alone.
> 2) To the best of our knowledge, both the conversion of dynamically sparsely trained ANN to SNN, and the relationship between firing-rate saturation and accuracy saturation in sparse SNN have not been quantitatively analyzed in prior SNN conversion work. Our study reveals (i) a consistent positive time lag across methods and architectures, and (ii) a significant difference in this time lag between sparse and dense networks. This provides new insight into how sparse connectivity affects temporal information processing in converted SNNs, which might be a potential cause for the performance and energy gap between networks with different sparsity level.
>
> To address the Reviewer concern:
> We have revised the text  of the article to include the discussion of this valuable point raised by the Reviewer. The text in the discussion highlighted that the competitive trade-off between accuracy and theoretical energy is not obtainable by either ANN-to-SNN conversion or CHT alone.
>
> **W2**: Energy claims are theoretical and hinge on strong assumptions; no hardware validation. The headline “up to 99% energy reduction” is derived from a spike-count/FLOP model plus constants (EMAC/EAC) and the assumption that sparse hardware gives linear speed/energy benefits w.r.t. link sparsity. There is no measurement on neuromorphic or sparse-compute hardware. Consequently, the core claim remains speculative without device-level latency/energy evidence or even cycle-accurate simulators.
>
> **Q1**: Hardware validation: Can you provide on-device latency and energy results for one platform to substantiate the 99% savings, and report how close real savings are to the theoretical model?
>
> **REPLY TO W2&Q1**:
>
> Thank you for your suggestion, we agree we should temperate our statement on energy savings, and state this limitation on energy saving tests in Discussion session.
>
> For this reason:
> 1) We modified title to **Investigating the Trade-off Between Accuracy and Theoretical Energy in Sparse ANN-to-SNN Conversion**  to elucidate that the energy savings here are theoretical.
> 2) In Discussion section, we state that limited by available hardware, we analyze theoretical energy consumption rather than measuring real energy consumption, which is a limitation of this work.
>
> **SEE FURTHER REPLIES IN NEXT COMMENTS**

---

> > ### Author Response · Authors · 2025-12-03
> >
> > **Q2**: Scope expansion: Do results persist on larger datasets (e.g., Tiny-ImageNet/ImageNet-subset) and deeper/backbone variants (e.g., ResNet/transformers) to support broad claims?
> >
> > **REPLY TO Q2**:
> >
> > Thank you for your deep insight. Following your suggestion, to demonstrate the generalizability of the method proposed here, we further conduct experiments on Vision Transformer (ViT). We proposed a hybrid structural-functional re-training pipeline for sparsifying the dense ViT model. In brief: we took the already dense Vision Transformer-Base (ViT-B, 16×16 patch size, 224×224 input) model proposed by Dosovitskiy et al[1] pre-trained for ImageNet classification task; we applied one-shot absolute weight pruning at 70% sparsity followed by CHT re-training; we converted the sparse re-trained model to SNN to evaluate its performance on the challenging ImageNet classification task. The last row of [Figure 2](https://anonymous.4open.science/r/rebuttal-figure-table-5424/Figure2.png) shows the accuracy of dense and sparse ANN ViTs and how SNN ViTs’ accuracy evolve with inference time step. Here the black/red numbers near the left y-axis report the dense/sparse ANN accuracy; the black/red numbers at the right report the maximal accuracy of dense/sparse SNNs. It could be observed that:
> >
> > (1) Sparse ANN ViT trained with CHT achieves comparable performance with dense ANN ViT
> >
> > (2) After conversion, both sparse and dense SNNs achieves similar accuracy.
> >
> > Furthermore, the last row in [Table 1](https://anonymous.4open.science/r/rebuttal-figure-table-5424/Table1.png)  compares the accuracy and theoretical energy consumption between sparse and dense SNN ViTs. It could be observed that:
> >
> > (1) The sparse SNN ViT can reduce 58.87% theoretical energy consumption compared to its dense counterpart with only 0.48% loss in accuracy.
> >
> > These results confirm that the competitive trade-off between accuracy and theoretical energy achieved by our method can also generalize to more complex network structures and more challenging tasks.
> >
> > [1] Dosovitskiy, Alexey. "An image is worth 16x16 words: Transformers for image recognition at scale." arXiv preprint arXiv:2010.11929 (2020).

---

### Official Review · Reviewer_FQAr · 2025-10-31

**Soundness:** 3
**Presentation:** 3
**Contribution:** 2
**Rating:** 2
**Confidence:** 4

**Summary:**

This paper explores a novel angle for ANN-to-SNN conversion: using dynamically sparsely trained ANNs as the source models. The authors employ Cannistraci-Hebb Training (CHT), a brain-inspired sparse training algorithm, to introduce sparsity into the ANN before conversion. The central claim is that this approach can produce sparse SNNs that achieve high energy efficiency (up to 99% savings compared to dense SNNs) while maintaining accuracy, representing a step towards more brain-like efficient computing.

**Strengths:**

The core idea of investigating dynamically sparse ANNs for conversion is innovative and represents a fresh contribution to the ANN2SNN field.

The focus on sparsity is well-aligned with the fundamental advantages of SNNs for energy-efficient, event-driven computation.

The method demonstrates good performance and significant energy savings on MLP (Multi-Layer Perceptron) architectures, validating the potential of the approach on simpler networks.

**Weaknesses:**

The title, "CONVERSION OF SPARSE ARTIFICIAL NEURAL NETWORK TO SPARSE SPIKING NEURAL NETWORK CAN SAVE UP TO 99% OF ENERGY," is potentially misleading. It suggests a 99% saving over a baseline that is not clearly specified, likely leading readers to assume it's compared to a sparse ANN. The abstract clarifies it's versus a dense SNN, but the title remains overly broad and risks overstating the finding.

The experiments are conducted on small datasets. The paper's impact would be significantly greater with validation on larger, more complex datasets (e.g., ImageNet or its subsets).

Table 2 reports sparsity and energy but crucially ​​omits the accuracy/performance​​ of the converted models. This makes it impossible to evaluate the true trade-off between efficiency and accuracy, which is the central claim of the work.

A key component of the method, CHT, is based on a preprint that has not undergone peer review. This reliance weakens the methodological foundation of the paper, as the core algorithm's efficacy and claims are not yet independently verified.

The discussion and conclusion frame the work as a significant step towards brain-like architecture. However, simply converting a sparsely trained ANN to an SNN is a relatively indirect contribution to neuromorphic computing. The narrative should be tempered to more accurately reflect the specific contribution: an efficient conversion pipeline leveraging sparse training, rather than a fundamental advance in brain-like computing.

**Questions:**

What are the accuracy results corresponding to the models in Table 2? Without these, the claim of high efficiency is incomplete. How much accuracy is sacrificed for the gained sparsity and energy savings?

Can the demonstrated benefits of this conversion approach scale to larger, modern datasets and architectures (e.g., deep convolutional networks)? What are the potential challenges?

Given that CHT itself is not a contribution of this paper and is not yet peer-reviewed, to what extent are the observed benefits specific to CHT versus being a general property of any high-quality sparse training method? Could similar results be achieved with other established sparse training techniques?

The 99% energy saving is compared to a dense SNN. What is the energy saving compared to a sparse SNN converted from a standard (non-CHT) pruned ANN? This would better isolate the contribution of the dynamic sparse training method.

---

> ### Author Response · Authors · 2025-12-03
>
> **NOTES FROM AUTHORS**
>
> 1) We have addressed the first major Reviewers’ concern on hyperparameter search. Now, the methodology for training dense ANN models is revised so that for dense and sparse ANNs, hyperparameter grid-search is conducted separately to find the best-performing models, instead of applying the best hyperparameters derived from sparse models on both dense and sparse networks.
> 2) We have addressed the second major Reviewers’ concern on computation of energy consumption in SNNs. Now, the energy is calculated by summing up the energy consumptions of all spikes on the synapses, instead of estimating energy based on average firing rates.
>
> **W1**: The title, "CONVERSION OF SPARSE ARTIFICIAL NEURAL NETWORK TO SPARSE SPIKING NEURAL NETWORK CAN SAVE UP TO 99% OF ENERGY," is potentially misleading. It suggests a 99% saving over a baseline that is not clearly specified, likely leading readers to assume it's compared to a sparse ANN. The abstract clarifies it's versus a dense SNN, but the title remains overly broad and risks overstating the finding.
>
> **REPLY TO W1**: Thank you for your kind advice. We have changed our title here to: **Investigating the Trade-off Between Accuracy and Theoretical Energy in Sparse ANN-to-SNN Conversion** to offer an accurate statement on our contribution
>
> **W2**: The experiments are conducted on small datasets. The paper's impact would be significantly greater with validation on larger, more complex datasets (e.g., ImageNet or its subsets).
>
> **REPLY TO W2**:
> Thank you for your deep insight. Following your suggestion, to demonstrate the generalizability of the method proposed here, we further conduct experiments on Vision Transformer (ViT). We proposed a hybrid structural-functional re-training pipeline for sparsifying the dense ViT model. In brief: we took the already dense Vision Transformer-Base (ViT-B, 16×16 patch size, 224×224 input) model proposed by Dosovitskiy et al[1] pre-trained for ImageNet classification task; we applied one-shot absolute weight pruning at 70% sparsity followed by CHT re-training; we converted the sparse re-trained model to SNN to evaluate its performance on the challenging ImageNet classification task. The last row of [Figure 2](https://anonymous.4open.science/r/rebuttal-figure-table-5424/Figure2.png) shows the accuracy of dense and sparse ANN ViTs and how SNN ViTs’ accuracy evolve with inference time step. Here the black/red numbers near the left y-axis report the dense/sparse ANN accuracy; the black/red numbers at the right report the maximal accuracy of dense/sparse SNNs. It could be observed that:
>
> (1) Sparse ANN ViT trained with CHT achieves comparable performance with dense ANN ViT
>
> (2) After conversion, both sparse and dense SNNs achieves similar accuracy.
>
> Furthermore, the last row in
> [Table 1](https://anonymous.4open.science/r/rebuttal-figure-table-5424/Table1.png)  compares the accuracy and theoretical energy consumption between sparse and dense SNN ViTs. It could be observed that:
>
> (1) The sparse SNN ViT can reduce 58.87% theoretical energy consumption compared to its dense counterpart with only 0.48% loss in accuracy.
>
> These results confirm that the competitive trade-off between accuracy and theoretical energy achieved by our method can also generalize to more complex network structures and more challenging tasks.
>
> [1] Dosovitskiy, Alexey. "An image is worth 16x16 words: Transformers for image recognition at scale." arXiv preprint arXiv:2010.11929 (2020).
>
> **W3**: Table 2 reports sparsity and energy but crucially ​​omits the accuracy/performance​​ of the converted models. This makes it impossible to evaluate the true trade-off between efficiency and accuracy, which is the central claim of the work.
>
> **REPLY TO W3**:
> Thank you for your kind suggestion. We have restructured [Table 1]((https://anonymous.4open.science/r/rebuttal-figure-table-5424/Table1.png) ) (previously Table 2)  to show not only the energy reduction but also the improvement in accuracy achieved by sparse SNNs compared to their dense counterparts. As can be seen, in 8 out of 13 groups of experiments, sparse SNNs can outperform the dense ones in accuracy, and the performances are comparable in other cases. And in all cases, sparse network can achieve a more than 30% reduction in energy, indicating a good trade-off between accuracy and theoretical energy
>
> **SEE FURTHER REPLIES IN NEXT  COMMENTS**

---

> > ### Author Response · Authors · 2025-12-03
> >
> > **W4**: A key component of the method, CHT, is based on a preprint that has not undergone peer review. This reliance weakens the methodological foundation of the paper, as the core algorithm's efficacy and claims are not yet independently verified.
> >
> > **REPLY TO W4**:
> > Thank you for raising your concern. Now the 2 CHT articles are published under double-blind peer review:
> >
> > (1)	The CHT method used for MLP and Transformer experiments is now published on NeurIPS 2025 main conference
> > Zhang, Yingtao, et al. "Brain network science modelling of sparse neural networks enables Transformers and LLMs to perform as fully connected." The Thirty-Ninth Annual Conference on Neural Information Processing Systems (2025). Accessible: https://openreview.net/forum?id=n4YLtNUsOJ
> >
> > (2)	CHT method for CNN is now available on NeurIPS workshop 2025 (which encompasses double-blinded peer review process): Li, Hanming, et al. "Cannistraci-Hebb Training of Convolutional Neural Networks." NeurIPS 2025 Workshop on Symmetry and Geometry in Neural Representations. Accessible: https://openreview.net/forum?id=CbuCWpDrIF
> >
> > **W5**: The discussion and conclusion frame the work as a significant step towards brain-like architecture. However, simply converting a sparsely trained ANN to an SNN is a relatively indirect contribution to neuromorphic computing. The narrative should be tempered to more accurately reflect the specific contribution: an efficient conversion pipeline leveraging sparse training, rather than a fundamental advance in brain-like computing.
> >
> > **REPLY TO W5**:
> > Thanks for your deep insight. We have now tempered our statement about the contribution of this article
> >
> > 1)	In Abstract:
> > < Together, these results demonstrate that Cannistraci-Hebb Training can be effectively integrated into ANN-to-SNN conversion pipelines to obtain sparse SNNs with competitive trade-off between accuracy and theoretical energy.>
> >
> > 2)	In Introduction:
> > < Together, these investigations position sparse ANNs derived from Cannistraci-Hebb Training as a useful component in developing efficient sparse ANN-to-SNN conversion pipelines, offering both practical computational benefits and a clearer understanding of the temporal dynamics of converted dense and sparse SNNs>
> >
> > 3)	In Discussion:
> > < By transferring the sparse advantage of Cannistraci-Hebb Training into spiking neural networks, this work achieves a competitive trade-off between accuracy and theoretical energy in sparse SNNs >
> >
> > **Q1**: What are the accuracy results corresponding to the models in Table 2? Without these, the claim of high efficiency is incomplete. How much accuracy is sacrificed for the gained sparsity and energy savings?
> >
> > **REPLY TO Q1**:
> >
> > Thanks for raising this question.
> > 1) We have restructured [Table 1](https://anonymous.4open.science/r/rebuttal-figure-table-5424/Table1.png) (previously Table 2) to show not only the energy reduction but also the improvement in accuracy achieved by sparse SNNs compared to their dense counterparts. As can be seen, in 8 out of 13 groups of experiments, sparse SNNs can outperform the dense ones in accuracy, and the performances are comparable in other cases. And in all cases, sparse network can achieve a more than 30% reduction in energy, indicating a good trade-off between accuracy and theoretical energy
> > 2) Regarding the request whether accuracy is sacrificed for energy saving, actually it is the opposite, sparsity can improve performance as can be seen in [Table 1](https://anonymous.4open.science/r/rebuttal-figure-table-5424/Table1.png) (previously Table 2) . The reason is that during CHT training, various topological properties critical to efficiency of a network such as low characteristic path length and hyperbolic community structure start to emerge as shown in Zhang et al[2] .  Also, sparsity in networks adds more non-linearity in learning, thus enabling the model to learn a better representation of features. We have now added this analysis to the Discussion session.
> >
> > [2] Zhang, Yingtao, et al. "Epitopological learning and cannistraci-hebb network shape intelligence brain-inspired theory for ultra-sparse advantage in deep learning." The Twelfth International Conference on Learning Representations. 2024.
> >
> > **SEE FURTHER REPLIES IN NEXT COMMENTS**

---

> ### Author Response · Authors · 2025-12-03
>
> **Q2**: Can the demonstrated benefits of this conversion approach scale to larger, modern datasets and architectures (e.g., deep convolutional networks)? What are the potential challenges?
>
> **REPLY TO Q2**:
>
> Thank you for your deep insight. Following your suggestion, to demonstrate the generalizability of the method proposed here, we further conduct experiments on Vision Transformer (ViT). We proposed a hybrid structural-functional re-training pipeline for sparsifying the dense ViT model. In brief: we took the already dense Vision Transformer-Base (ViT-B, 16×16 patch size, 224×224 input) model proposed by Dosovitskiy et al[3] pre-trained for ImageNet classification task; we applied one-shot absolute weight pruning at 70% sparsity followed by CHT re-training; we converted the sparse re-trained model to SNN to evaluate its performance on the challenging ImageNet classification task. The last row of [Figure 2](https://anonymous.4open.science/r/rebuttal-figure-table-5424/Figure2.png) shows the accuracy of dense and sparse ANN ViTs and how SNN ViTs’ accuracy evolve with inference time step. Here the black/red numbers near the left y-axis report the dense/sparse ANN accuracy; the black/red numbers at the right report the maximal accuracy of dense/sparse SNNs. It could be observed that:
>
> (1)	Sparse ANN ViT trained with CHT achieves comparable performance with dense ANN ViT
>
> (2)	After conversion, both sparse and dense SNNs achieves similar accuracy.
>
> Furthermore, the last row in
> [Table 1](https://anonymous.4open.science/r/rebuttal-figure-table-5424/Table1.png) compares the accuracy and theoretical energy consumption between sparse and dense SNN ViTs. It could be observed that:
>
> (1) The sparse SNN ViT can reduce 58.87% theoretical energy consumption compared to its dense counterpart with only 0.48% loss in accuracy.
>
> These results confirm that the competitive trade-off between accuracy and theoretical energy achieved by our method can also generalize to more complex network structures and more challenging tasks.
>
> [3] Dosovitskiy, Alexey. "An image is worth 16x16 words: Transformers for image recognition at scale." arXiv preprint arXiv:2010.11929 (2020).
>
> **Q3**: Given that CHT itself is not a contribution of this paper and is not yet peer-reviewed, to what extent are the observed benefits specific to CHT versus being a general property of any high-quality sparse training method? Could similar results be achieved with other established sparse training techniques?
>
> **Q4**: The 99% energy saving is compared to a dense SNN. What is the energy saving compared to a sparse SNN converted from a standard (non-CHT) pruned ANN? This would better isolate the contribution of the dynamic sparse training method.
>
> **REPLY TO Q3&Q4**:
>
> Thank you for your valuable suggestion. We have added a new section in Appendix named [CHT vs. Pruning Under Matched Sparsity]. In this section, we compare the network-wise sparsity, energy and accuracy of 2 SNN VGG-16 networks derived from CHT and an influential structural pruning method DepGraph[4] with similar sparsity, converted with QCFS method (Method 1), which is widely adopted. Looking at [Table A2](https://anonymous.4open.science/r/rebuttal-figure-table-5424/TableA2.png), the results suggest that across 2 different datasets CIFAR10 and CIFAR100, under similar sparsity, CHT has higher accuracy while achieving comparable energy consumption. This is especially obvious in CIFAR100, where CHT’s accuracy is much higher than DepGraph (74.32% vs. 71.16%).
>
> Furthermore, we compare the inference and energy consumption between sparse SNNs directly trained from STBP and converted from CHT-trained sparse ANNs, we selected 2 state-of-the-art STBP[5]-based SNN sparse training methods to compare with CHT, namely SD-SNN[6] and DPAP[7]. We adopt the same network structure as in SOTA articles, which is a convolutional network with 6 convolutional layers followed by 2 fully connected layers, and do dynamic sparse training on CHT to derive the sparse ANN, which is then converted to SNN with QCFS (Method 1) for comparison.
>
> [Table A3](https://anonymous.4open.science/r/rebuttal-figure-table-5424/TableA3.png) shows a comparison on SNN accuracy and energy consumptions between different methods. The results suggest that across 2 CIFAR datasets, CHT networks achieves the highest network-wise sparsity (87.1%). Given same inference time steps (T=8), CHT consumes least energy and achieves comparable accuracies with STBP-based sparse training methods. We now add this as a new section [CHT vs. STBP-Based SNN sparse training Methods] in the appendix.
> Thus, according to the results above, the trade-off between accuracy and theoretical energy does not derive from mere sparsity level, but also depends on the method to achieve sparsity, illustrating the role of dynamic sparse training method such as CHT.
>
> **SEE FURTHER REPLIES IN NEXT COMMENTS**

---

> > ### Author Response · Authors · 2025-12-03
> >
> > [4] Fang, Gongfan, et al. "Depgraph: Towards any structural pruning." Proceedings of the IEEE/CVF conference on computer vision and pattern recognition. 2023.
> >
> > [5] Wu, Yujie, et al. "Spatio-temporal backpropagation for training high-performance spiking neural networks." Frontiers in neuroscience 12 (2018): 331.
> >
> > [6] Han, Bing, et al. "Adaptive sparse structure development with pruning and regeneration for spiking neural networks." Information Sciences 689 (2025): 121481.
> >
> > [7] Han, Bing, et al. "Developmental plasticity-inspired adaptive pruning for deep spiking and artificial neural networks." IEEE Transactions on Pattern Analysis and Machine Intelligence (2024).

---

### Official Review · Reviewer_NFnN · 2025-11-01

**Soundness:** 2
**Presentation:** 2
**Contribution:** 2
**Rating:** 2
**Confidence:** 5

**Summary:**

This paper proposes a novel and promising approach that combines Dynamic Sparse Training with ANN-to-SNN conversion. The authors employ Cannistraci-Hebb Training to train highly sparse ANNs and successfully convert them into sparse SNNs. The results demonstrate that these sparse SNNs can achieve accuracy comparable to or even surpassing their dense counterparts, while achieving a remarkable theoretical energy reduction of up to 99%. Furthermore, the paper is the first to reveal the phenomenon of a time lag between the saturation of firing rate and accuracy in SNNs, and finds a significant difference in this lag between sparse and dense networks, providing new insights into the underlying mechanisms of SNNs. Overall, this is an important work that synergizes the advantages of structural sparsity and temporal sparsity.

**Strengths:**

1.This is a study on converting dynamically sparsely trained ANNs into SNNs, while prior work has mostly focused on converting dense networks.

2.The authors validate their findings across two different network architectures (MLP and VGG-16), two datasets (CIFAR-10/100), and three representative conversion methods.

**Weaknesses:**

1.The experiments are conducted solely on traditional CNNs like MLP and VGG-16. Currently, Transformer architectures have become mainstream in fields such as computer vision. To demonstrate the generalizability and state-of-the-art relevance of the proposed method, the authors should include experimental results on converting sparsely trained Transformer models from ANN to SNN.

2.All experiments are performed on the relatively small CIFAR-10 and CIFAR-100 datasets. The absence of validation on large-scale, more challenging real-world datasets like ImageNet raises concerns about the generalization capability of the conclusions in complex scenarios and diminishes the practical value of the method.

**Questions:**

1.Regarding the relatively small energy improvement (only 19%) for VGG-16 under the AEC method, the paper attributes it to the sparse SNN requiring a longer inference time T. Could the authors analyze why, under the AEC method, the 50%-sparse VGG-16 requires a longer T to reach peak accuracy? Does this suggest a potential incompatibility between certain conversion methods and sparse topologies?

---

> ### Author Response · Authors · 2025-12-03
>
> **NOTES FROM AUTHORS**
>
> 1) We have addressed the first major Reviewers’ concern on hyperparameter search. Now, the methodology for training dense ANN models is revised so that for dense and sparse ANNs, hyperparameter grid-search is conducted separately to find the best-performing models, instead of applying the best hyperparameters derived from sparse models on both dense and sparse networks.
> 2) We have addressed the second major Reviewers’ concern on computation of energy consumption in SNNs. Now, the energy is calculated by summing up the energy consumptions of all spikes on the synapses, instead of estimating energy based on average firing rates.
>
> **W1**:The experiments are conducted solely on traditional CNNs like MLP and VGG-16. Currently, Transformer architectures have become mainstream in fields such as computer vision. To demonstrate the generalizability and state-of-the-art relevance of the proposed method, the authors should include experimental results on converting sparsely trained Transformer models from ANN to SNN.
>
> **W2**:All experiments are performed on the relatively small CIFAR-10 and CIFAR-100 datasets. The absence of validation on large-scale, more challenging real-world datasets like ImageNet raises concerns about the generalization capability of the conclusions in complex scenarios and diminishes the practical value of the method.
>
> **REPLY FOR W1&W2**:
> Thank you for your deep insight. Following your suggestion, to demonstrate the generalizability of the method proposed here, we further conduct experiments on Vision Transformer (ViT). We proposed a hybrid structural-functional re-training pipeline for sparsifying the dense ViT model. In brief: we took the already dense Vision Transformer-Base (ViT-B, 16×16 patch size, 224×224 input) model proposed by Dosovitskiy et al[1] pre-trained for ImageNet classification task; we applied one-shot absolute weight pruning at 70% sparsity followed by CHT re-training; we converted the sparse re-trained model to SNN to evaluate its performance on the challenging ImageNet classification task. The last row of [Figure 2](https://anonymous.4open.science/r/rebuttal-figure-table-5424/Figure2.png) shows the accuracy of dense and sparse ANN ViTs and how SNN ViTs’ accuracy evolve with inference time step. Here the black/red numbers near the left y-axis report the dense/sparse ANN accuracy; the black/red numbers at the right report the maximal accuracy of dense/sparse SNNs. It could be observed that:
>
> (1)	Sparse ANN ViT trained with CHT achieves comparable performance with dense ANN ViT
>
> (2)	After conversion, both sparse and dense SNNs achieves similar accuracy.
>
> Furthermore, the last row in
> [Table 1](https://anonymous.4open.science/r/rebuttal-figure-table-5424/Table1.png)  compares the accuracy and theoretical energy consumption between sparse and dense SNN ViTs. It could be observed that:
>
> (1)	 The sparse SNN ViT can reduce 58.87% theoretical energy consumption compared to its dense counterpart with only 0.48% loss in accuracy.
>
> These results confirm that the accuracy/theoretical-energy trade-off achieved by our method can also generalize to more complex network structures and more challenging tasks.
>
> [1] Dosovitskiy, Alexey. "An image is worth 16x16 words: Transformers for image recognition at scale." arXiv preprint arXiv:2010.11929 (2020).
>
> **SEE FURTHER REPLIES IN NEXT  COMMENTS**

---

> > ### Author Response · Authors · 2025-12-03
> >
> > **Q1**: Regarding the relatively small energy improvement (only 19%) for VGG-16 under the AEC method, the paper attributes it to the sparse SNN requiring a longer inference time T. Could the authors analyze why, under the AEC method, the 50%-sparse VGG-16 requires a longer T to reach peak accuracy? Does this suggest a potential incompatibility between certain conversion methods and sparse topologies?
> >
> > **REPLY TO Q1**:
> > 1) As mentioned above, the methods for training dense ANN models and calculating the energy consumption are now revised. After updating the results, it could be found in [Table 1](https://anonymous.4open.science/r/rebuttal-figure-table-5424/Table1.png) in the article that the energy reduction now becomes 31.79% instead of 19% for SNN VGG-16 converted by QCFS method (tested on CIFAR10)
> > 2) Regarding the reason for long inference time in AEC method (method 3), in the case of AEC, the definition of T is different from QCFS (method 1) and SNM (method 2). Instead of performing integrate-and-fire operation in real-time, AEC do integrate and fire operation separately in two time window, and in the experiments a detailed grid-search is performed on this window size (within {2,4,8,16,32,64}) and the best-performing window size is denoted here as T as stated in the caption of article’s [Figure2](https://anonymous.4open.science/r/rebuttal-figure-table-5424/Figure2.png). Under this circumstance, a larger window size will allow longer integration-and-fire time for the model to process more spiking information, thus resulting in higher accuracy. Referring to Figure 2 in the article, this trend can be observed in both dense and sparse models, so this phenomenon is most likely related to the methodology design in AEC.
> >
> > We have revised the text  of the article to include the discussion of this valuable point raised by the Reviewer in the Discussion section

---

### Author Response · Authors · 2025-12-03

# Summary

We sincerely thank all reviewers for their valuable time and insightful feedback, which is very helpful in further improving the quality of our paper.

We appreciate that the reviewers recognized: (1) the novelty of exploring the advantages of converting dynamically sparse trained ANNs into SNNs (FQAr, nyYp) (2) the relevance of combining sparsity with conversion-based SNNs for energy-efficient neuromorphic computing (NFnN, FQAr, nyYp) (3) the usefulness of the observed firing–accuracy time-lag phenomenon in understanding sparse vs. dense SNN behavior (NFnN) (4) the clear experimental pipeline and readability of the manuscript (nyYp). We are also encouraged by the positive assessments that our motivation is timely and grounded (nyYp), the approach and results are novel and promising (FQAr, NFnN).

Nevertheless, our initial scores were all low (rating 2) particularly because we did not provide experiments on larger datasets, a request that we fully addressed in this revision.

# Additional Experiments

1) **Experiments on more challenging tasks and modern architectures**

    To show that the claimed trade-off between accuracy and theoretical energy achieved by our method can generalize to larger-scale dataset and more up-to-date network architectures, we added extensive new simulations on ImageNet using Vision Transformer (ViT-B) sparsely re-trained with CHT and converted into SNN. The results show that 70% sparse SNN converted from sparse ViT can reduce 58% energy consumption while achieving comparable accuracy compared to its dense counterpart. These results directly address concerns regarding scalability and generality (Reviewers NFnN, FQAr, nyYp, 3cJ8).

2) **Comparison with sparse ANN baselines beyond CHT**

    To show that CHT provides unique sparsity benefits to sparse SNNs converted from sparse ANNs, we compare sparse SNNs converted from CHT-trained sparse ANNs and pruned sparse ANNs (using influential pruning method DepGraph) under similar sparsity. Results show that the sparse SNNs derived from CHT-based ANNs can achieve higher accuracy while requiring comparable energy consumption with respect to those derived from DepGraph pruning methods. (Reviewer FQAr)

3) **Comparison with SNN sparse-training baselines**

    To provide a comparison between sparse SNNs derived from CHT-trained ANNs, and from STBP-based direct SNN sparse training, 2 state-of-the-art SNN sparse training methods SD-SNN and DPAP are employed. Being tested on the same network architecture, results show that sparse SNNs converted from CHT-trained ANNs can achieve much higher sparsity, lower energy consumption and comparable accuracy on image classification tasks CIFAR10 and CIFAR100. (Reviewer 3cJ8)

# Clarifications and Methodological Fixes

1) Comparing energy consumption and accuracy between dense and sparse SNNs jointly
To illustrate that sparse SNNs can not only reduce energy consumption, but also retain accuracy compared to its dense counterpart, Table 1 now shows the energy reduction and accuracy improvement on sparse SNNs simultaneously. As can be seen, in 8 out of 13 experiments sparse SNNs can achieve superior accuracy, and in all experiments achieve more than 30% energy reduction compared to dense SNNs. These results address concerns on lack of accuracy comparison between dense and sparse spiking models (Reviewers FQAr, 3cJ8)

2) Corrected and clarified energy computation.
We revised the computation of energy consumption to sum up the energy required of all spikes on synapses instead of estimating energy according to average firing rate.

3) Corrected ANN training hyper-parameter grid-search
We revised the hyper-parameter search on training ANNs, where best hyper-parameter grid-search is now performed on dense and sparse ANNs separately, instead of using the best hyper-parameters on sparse ANNs to both dense and sparse SNNs.

4) Detailed explanation of why different ANNtoSNN conversion method requires different T
For Reviewer NFnN’s question on increased T under AEC conversion method, we provide a clarification that the definition of T varies across ANNtoSNN conversion methods, and why a higher T is observed specific to AEC (Reviewer NFnN)

5) Tempered and clarified impact narrative
We improved the title, abstract, introduction and discussion session to provide a clearer and fairer statement on our contribution (Reviewers FQAr, nyYP)

6) Formatting, tables, and figure layout
Figures and tables in the article are reorganized to provide a clearer presentation on the results, and improved readability (Reviewer 3cJ8)

**CONTINUED IN NEXT COMMENT**

---

> ### Author Response · Authors · 2025-12-03
>
> We hope that these additions and clarifications address the reviewers’ comments and improve the precision, completeness, and fairness of our claims. We believe the revised submission clearly communicates both the strengths and the limitations of our method and provides a more robust and realistic evaluation of its potential. Despite we got an initial score of 2 from all reviewers, the questions raised by the Reviewers were clear and manageable to reply. We conducted a meticulous revision that addressed all Reviewers’ concerns, and we hope the Area Chair will recognize this merit and promote our article.

---

### Meta-Review · Area_Chair_sNHk · 2026-01-01

**Summary:**

Based on the reviewers' evaluations and the authors' subsequent revisions, this paper proposes a method for converting dynamically sparse-trained Artificial Neural Networks (ANNs) into sparse Spiking Neural Networks (SNNs) to enhance energy efficiency. While the reviewers acknowledged the timeliness of the topic and some novel aspects of the investigation, significant concerns regarding the paper's contribution, validation, and presentation led to a consensus recommendation for rejection. The primary concerns are summarized as follows:

Limited Novelty and Incremental Contribution: Multiple reviewers (nyYp, 3cJ8) noted that the work primarily combines two established techniques—Dynamic Sparse Training (DST) and ANN-to-SNN conversion—without introducing a fundamental methodological advance. The core contribution was perceived as an empirical evaluation of this combination rather than a novel algorithmic or theoretical innovation for the SNN community.

Insufficient Empirical Validation and Generalizability: Initially, all reviewers criticized the narrow experimental scope, which was limited to small datasets (CIFAR-10/100) and traditional architectures (MLP, VGG-16). Although the authors added experiments with Vision Transformers (ViT) on ImageNet in their revision, concerns remained about whether the demonstrated benefits—particularly the headline claim of "up to 99% energy savings"—are robust, scalable, and generalizable to state-of-the-art models and real-world applications.

Overstated Claims and Lack of Hardware Verification: Reviewers (FQAr, nyYp) found the original title and energy-saving claims to be potentially misleading, as they were based on theoretical calculations rather than actual hardware measurements. The absence of on-device validation undermines the practical relevance of the energy efficiency claims, which are central to the paper's impact.

Incomplete Comparison and Evaluation: Reviewers (FQAr, 3cJ8) pointed out that initial results lacked critical comparisons, such as accuracy trade-offs for sparse models and benchmarks against established sparse SNN training methods (e.g., STBP-based approaches). While the authors added comparisons in the revision, questions persisted about whether the advantages of the proposed pipeline are unique to the Cannistraci-Hebb Training (CHT) method or are general properties of high-quality sparse training.

Presentation and Clarity Issues: Several reviewers noted problems with the organization of tables, figures, and the overall narrative. Although the authors attempted to improve clarity in the rebuttal, the initial presentation was deemed suboptimal, affecting the readability and professionalism of the submission.

Overall Decision: Reject

Despite the authors' efforts to address reviewers' concerns through additional experiments and clarifications, the paper was ultimately assessed as lacking sufficient novelty, rigorous validation, and convincing evidence of a significant contribution to the field. The revised submission did not fully overcome the fundamental issues regarding its incremental nature, overstated claims, and limited practical verification.

**Reviewer Concerns:**

Concerns Addressed by the Rebuttal:

Improved Presentation and Clarity:

Reviewers' Concerns (FQAr, 3cJ8): Cited issues with a misleading title, omitted accuracy data in tables, and poor figure/table layout.

Authors' Response: Effectively addressed by:

Changing the title to be more precise ("Investigating the Trade-off Between Accuracy and Theoretical Energy...").

Restructuring tables (e.g., new Table 1) to jointly present accuracy and energy reduction.

Reorganizing figures and supplementary materials for better readability.

Provision of Requested Comparisons:

Reviewers' Concerns (FQAr, 3cJ8): Asked for comparisons with other sparse training methods (pruning, direct SNN training).

Authors' Response: Directly addressed by adding two new comparative analyses in the appendix:

CHT vs. Pruning (DepGraph): Showing CHT leads to higher accuracy under similar sparsity.

CHT-based Conversion vs. STBP-based Sparse SNN Training (SD-SNN, DPAP): Showing competitive accuracy with higher sparsity and lower energy at comparable time steps.

Outstanding Concerns (Fundamental to Rejection):
Lack of Compelling Novelty / Incremental Contribution:

Reviewers' Stance (nyYp, 3cJ8): The core criticism that the work is a combination of established techniques rather than a novel methodological contribution was not overcome. The rebuttal reinforced that this is an evaluation of combining CHT (DST) with ANN2SNN conversion, not a new algorithm or theory for SNNs.

Theoretical vs. Practical Energy Claims:

Reviewers' Stance (nyYp, FQAr): While the authors acknowledged the limitation, the lack of hardware validation remains a critical, unaddressed gap. The claim of "up to 99% energy savings" is fundamentally weakened by being purely theoretical and model-based. For a paper whose major selling point is energy efficiency, this is a decisive weakness for a top-tier venue like ICLR.

Questionable Specificity of Benefits to CHT:

Reviewers' Stance (FQAr): Although comparisons were added, the rebuttal does not fully dispel the doubt that the observed benefits might be a general property of any high-quality sparse training method rather than something unique to CHT. The added comparisons show CHT is good, but not conclusively why this specific pipeline is a breakthrough.

Despite a strong rebuttal that addressed many specific critiques, the paper was deemed to lack the novelty, methodological innovation, and verifiable practical impact required for acceptance at ICLR 2026.

**Reviewer Scores:**

Reviewer NFnN (Initial Score: 2 - Reject)
Primary Concerns: Lack of experiments on large-scale datasets (ImageNet) and modern architectures (Transformers). Also sought clarification on the AEC conversion method's interaction with sparsity.

Reaction to Rebuttal: This reviewer's concerns were the most directly and fully addressed by the rebuttal. The authors added compelling ImageNet/ViT experiments and provided a clear technical explanation for the AEC timing issue. The reviewer's summary was initially positive ("novel and promising"), and the low score seemed primarily due to the lack of scalability experiments.

Estimated Score Change: 2 → 2 or 4.

Rationale: The new experiments directly satisfy the reviewer's two major "Weaknesses." The core idea, which the reviewer already found promising, is now backed by evidence on a state-of-the-art scale. It is likely this reviewer would have upgraded their score to 4 or maintained 2.

Reviewer FQAr (Initial Score: 2 - Reject)
Primary Concerns: Misleading title, small datasets, missing accuracy data in tables, reliance on an unpublished CHT method, and overstated narrative about brain-like computing.

Reaction to Rebuttal: The authors addressed the majority of this reviewer's specific, actionable points: they changed the title, added ImageNet/ViT experiments, restructured tables to include accuracy, provided peer-review citations for CHT, and tempered the narrative. They also added the requested comparisons with pruning and STBP methods.

Estimated Score Change: 2 → Likely 2, possibly 4.


Reviewer nyYp (Initial Score: 2)
Primary Concerns: Incremental novelty ("combines two well-established components") and lack of hardware validation for energy claims, making them speculative.

Reaction to Rebuttal: The authors partially addressed these concerns. They added experiments to improve generalizability (Q2) and tempered their claims by acknowledging the theoretical limitation. However, they could not change the fundamental nature of the contribution or provide hardware validation.

Estimated Score Change: 2 → 2 (Reject).

Rationale: This reviewer's core criticism was about novelty and speculative claims. The rebuttal does not transform the work into a novel methodological contribution; it remains a thorough evaluation of a combined pipeline. Furthermore, the lack of hardware validation, which the reviewer explicitly requested ("Can you provide on-device latency and energy results..."), remains an outstanding, critical weakness.

Reviewer 3cJ8 (Initial Score: 2 - Reject)
Primary Concerns: Missing accuracy data, small-scale experiments, no comparison to STBP-based SNN training, high inference time steps, and questioning the contribution to the SNN community.

Reaction to Rebuttal: The authors addressed many of the specific technical critiques: they added accuracy data (W1), ImageNet/ViT experiments (W1), and extensive comparisons to STBP methods and pruning (W2, W3). They also improved presentation (W5).

Estimated Score Change: 2 → 2 (Reject).

Rationale: Despite the strong technical response, this reviewer's most damning concern was the contribution to the SNN community (W4). The reviewer's perspective seems to be that significant SNN research should involve direct SNN training innovations. The authors' rebuttal frames the contribution as a pipeline and new analysis (firing-accuracy lag), but this may not satisfy a reviewer who values novel SNN training algorithms above conversion pipelines. The reviewer's confidence was a 5 ("absolutely certain"), indicating a strong, fixed position. They are the least likely to have changed their score.

---

### Decision · Program_Chairs · 2026-01-26

Reject